# Challenges in Treating Charcot-Marie-Tooth Disease and Related Neuropathies: Current Management and Future Perspectives

**DOI:** 10.3390/brainsci11111447

**Published:** 2021-10-29

**Authors:** Chiara Pisciotta, Paola Saveri, Davide Pareyson

**Affiliations:** Unit of Rare Neurodegenerative and Neurometabolic Diseases, Department of Clinical Neurosciences, Fondazione IRCCS Istituto Neurologico Carlo Besta, Via Celoria 11, 20133 Milan, Italy; chiara.pisciotta@istituto-besta.it (C.P.); paola.saveri@istituto-besta.it (P.S.)

**Keywords:** CMT, therapy, management, clinical trials, gene therapy, gene silencing, inherited neuropathy

## Abstract

There is still no effective drug treatment available for Charcot-Marie-Tooth neuropathies (CMT). Current management relies on rehabilitation therapy, surgery for skeletal deformities, and symptomatic treatment of pain; fatigue and cramps are frequent complaints that are difficult to treat. The challenge is to find disease-modifying therapies. Several approaches, including gene silencing, to counteract the *PMP22* gene overexpression in the most frequent CMT1A type are under investigation. PXT3003 is the compound in the most advanced phase for CMT1A, as a second-phase III trial is ongoing. Gene therapy to substitute defective genes or insert novel ones and compounds acting on pathways important for different CMT types are being developed and tested in animal models. Modulation of the Neuregulin pathway determining myelin thickness is promising for both hypo-demyelinating and hypermyelinating neuropathies; intervention on Unfolded Protein Response seems effective for rescuing misfolded myelin proteins such as P0 in CMT1B. HDAC6 inhibitors improved axonal transport and ameliorated phenotypes in different CMT models. Other potential therapeutic strategies include targeting macrophages, lipid metabolism, and Nav1.8 sodium channel in demyelinating CMT and the P2X7 receptor, which regulates calcium influx into Schwann cells, in CMT1A. Further approaches are aimed at correcting metabolic abnormalities, including the accumulation of sorbitol caused by biallelic mutations in the sorbitol dehydrogenase (*SORD*) gene and of neurotoxic glycosphingolipids in HSN1.

## 1. Introduction

Charcot-Marie-Tooth disease collects a genetically heterogeneous group of inherited disorders sharing a common core phenotype of length-dependent progressive sensory-motor or motor neuropathy with foot deformities and altered deep tendon reflexes. It is classified according to nerve conduction studies into a demyelinating and an axonal variety: CMT1 for demyelinating (with slowed nerve conduction velocities, NCVs) neuropathies with autosomal dominant inheritance, CMT4 for autosomal recessive demyelinating forms, and CMT2 for primary axonal types with autosomal dominant or recessive transmission. There are intermediate forms of CMT with NCV values falling in between CMT1 and CMT2: the most relevant is CMTX1, which is associated with mutations in the *GJB1* gene on chromosome Xq13.1; less common autosomal dominant or recessive forms are on the record. Pure motor forms of CMT are characterised by the preservation of sensory nerves and are labelled distal Hereditary Motor Neuropathies (dHMNs), which can have autosomal dominant, autosomal recessive, or X-linked transmission. Pure or predominantly sensory neuropathies are grouped under the term of Hereditary Sensory (and Autonomic) Neuropathies (HS(A)Ns) and can have autosomal dominant or recessive inheritance patterns. Eventually, Hereditary Neuropathy with liability to Pressure Palsies is an autosomal dominant disorder characterised by recurrent focal neuropathies, which are typically painless and transient, and it is associated with the deletion of the *PMP22* gene, whereas the duplication of the same gene with PMP22 overexpression is associated with the most frequent CMT subtype, CMT1A. Overall, this group of inherited peripheral neuropathies is also labelled as CMT and related neuropathies and is further classified according to the mutated gene. Indeed, about 100 genes have been associated with CMT and related disorders, which code for proteins involved in many different functions relevant for the nerves, including myelin formation and maintenance, axonal transport, vesicular trafficking, mitochondrial homeostasis, Endoplasmic Reticulum (ER) stress, channels formation, etc. [1,2].

Thanks to the advances in genetics and in cellular and animal models, our understanding of pathomechanisms of CMT and the genetic diagnostic yield have greatly increased during the last years. However, we are still awaiting an effective therapy that is able to modify the natural disease course of CMT and related neuropathies, and treatment is still based on symptomatic pharmacological treatment, surgery for skeletal deformities, and rehabilitation therapy. Nevertheless, there are many potential therapies in the pipeline that are generating a cautiously optimistic view among researchers in the field. In this review, we will examine the current management of patients with CMT and related disorders and analyse the perspectives for potential future treatments.

## 2. Current Management

### 2.1. Management of Pain, Fatigue, and Cramps

Pain is a frequent complaint in CMT (reported by 23–85% of patients) and may be biomechanical–nociceptive in nature, linked to skeletal deformities of the foot and spine, leading to altered posture and arthropathic degeneration or a true neuropathic pain [3,4,5,6,7,8,9,10]. It usually has mild-to-moderate intensity: the mean visual analogue scale (VAS) score for pain value in 272 CMT1A patients recruited in the ascorbic acid trial was 3.7 ± 3.0 on a 0–10 scale [11]. More than one-third of CMT patients make use of analgesic drugs [9], mainly non-steroid anti-inflammatory drugs (NSAIDs) and paracetamol/acetaminophen, and less commonly opioids [5]. About 18–30% of patients complain of neuropathic pain [4,6], which should be treated according to evidence-based medicine by choosing first-line and second-line treatments among tricyclic antidepressants, SSRI drugs, anticonvulsants (pregabalin, gabapentin, carbamazepine, etc.), and local capsaicin, avoiding the use of opioids [12,13].

Fatigue is also a common symptom for CMT patients [14] impacting on the quality of their life [15,16]. Modafinil showed some efficacy in a small pilot study on four patients, but it carries an unacceptable risk of side effects [17]; the usefulness of amantadine, tested in Guillain–Barrè syndrome, is not established. Muscle cramps are another frequent complaint, reported by about 85% of the patients [15], for which no treatment has shown clear efficacy including the commonly used magnesium supplementation [18]; quinine is the only drug that proved effective in previous studies for idiopathic leg cramps, but the risk of severe side effects outweighs the potential benefits [19]. Rehabilitation therapy with stretching may be advantageous for treating cramps and pain [20]. Oral FLX-787, an activator of TRPA1 and TRPV1 channels, was tested in a randomised, double-blind, placebo-controlled, phase II study (ClinicalTrials.gov NCT03254199); however, this was stopped because of oral intolerability in some patients.

### 2.2. Rehabilitation Therapy

Physical and occupational therapy is usually recommended on the basis of empirical experience and an increasing number of clinical studies. Recommended treatment protocols include muscle strengthening, aerobic conditioning, stretching, and exercise for posture and balance [21,22,23]. The first randomised clinical trials in the 1990s by Lindeman [24,25,26] and colleagues demonstrated that mild to moderate exercise in programs aimed at reinforcing lower limb proximal muscles is beneficial in CMT and resulted in increased strength as compared to placebo, whereas more intensive programs were less tolerated. In the first years of the novel millennium, two cohort studies confirmed that the strengthening of upper and lower proximal muscles and of handgrip is useful and well-tolerated [27,28]. El Mahdi et al. in 2008 showed that aerobic conditioning with cyclo-ergometer training improved cardiorespiratory parameters as compared to placebo [29]. Matjacic and Zupan tested a brief combination therapy with stretching, muscle strengthening, and balance training with significant improvement in balance [30]. A pilot study of lower limb proximal muscle strengthening in 26 CMT subjects gave a modest outcome as strength increased in left but not right hip flexors, and retrospective power analysis showed that more patients are needed in such trials [31]. Burns et al. performed a double-blind, sham-controlled trial in 60 children with CMT1A with progressive resistance exercise of foot dorsiflexors for 6 months: treated children had stronger foot dorsiflexion than controls at 6-month assessment (end of treatment), which was more evident at 12 months and was maintained at 24 months (although other parameters did not differ between treated and untreated groups) [32]. Wallace et al. tested, in a randomised single-blinded crossover trial on 23 CMT1A subjects, a 12-week aerobic training program consisting of 36 work sessions on a bicycle ergometer, which resulted in a significant improvement of Peak Oxygen Uptake (VO2 peak) [33]. Mori et al. compared two different 12-week protocols in 53 CMT1A patients, the first comprising stretching and proprioceptive exercise (SPe) and the second including an additional treadmill training (TreSPe): both groups significantly benefitted from treatments as far as walking and balance measures were concerned [34]. Pazzaglia et al. (2016) treated 14 CMT1A patients with focal mechanical vibration, which seems to improve the somatosensory system function, on quadriceps and triceps surae muscles for 3 days and noticed a significant effect on balance as assessed with the Berg Balance Scale and the Dynamic Gait Index [35].

Therefore, there is evidence that approaches including muscle strengthening and cardiorespiratory conditioning are beneficial for CMT patients. However, most trials tested only short treatment periods, usually of 8–12 weeks, and the benefit tends to be lost over time, underlining the need for continuous or repeated treatments.

Many CMT patients use shoe inserts to reduce foot misalignment, pain, and callosities determined by foot deformities and ankle foot orthoses (AFOs) or orthopaedic shoes to correct the foot-drop during walking and the ankle instability. AFOs are frequently poorly tolerated because of pressure sores, discomfort, or aesthetic reasons; there are now several different solutions, including customised AFOs and lighter models, that can be successfully employed to overcome such issues [36]. A minority of patients need aids for walking, such as canes, crutches, or walkers, or are confined to a wheelchair.

### 2.3. Surgical Treatment

About 20% of CMT patients undergo surgery for foot deformities, often with more than one intervention needed. Surgical intervention is aimed at re-aligning the foot in a proper position, correcting the muscle imbalance, and improving pain. Such goals are reached by acting on soft tissues (plantar fascia release, tendon transfers, tendon lengthening), bones (first metatarsal bone, calcaneal osteotomies, toe straightening) or joints (e.g., double arthrodesis) in a variable combination of procedures. A survey on current practice revealed that there were differences in the approaches between different centres and countries [37]. During the last few years, commendable efforts have been devoted to compare experiences and reach consensus on which procedures should be done and which should be avoided [38,39]. Surgery, if indicated ideally after multidisciplinary evaluation and in the hands of an orthopaedic surgeon with expertise in CMT, should occur early, as fixed deformities are more difficult to treat, in an individualised manner and according to determined sequences of procedures.

Scoliosis occurs in 20–30% of patients and may require bracing, physical therapy, and rarely surgery [40], according to unpublished data from the Italian CMT Registry.

On one hand, orthoses and tendon transfers can improve the pinch and grasp function. Nerve release or transfer to relieve entrapment neuropathies is often performed in HNPP patients before diagnosis, sometimes with benefit but more often without. The surgical decision in diagnosed patients should be individualised and well balanced after careful clinical, electrophysiological, and ultrasound evaluation. Most patients recover spontaneously from palsies, and therefore, surgery should be reserved for patients with persistent symptoms (particularly pain) related to entrapments.

### 2.4. General Care

The Internet contains too-long lists of drugs that are labelled as contraindicated for CMT patients. It should be borne in mind that the use of drugs that are toxic for the peripheral nerve should be avoided if possible, but that the decision depends on the duration of treatment (a short course is not a problem), the degree of toxicity (many drugs carry very low risks), and the risk–benefit ratio. The most difficult choices are related to patients with cancer, lymphoma, or leukaemia for whom vinka alkaloid, platin, or taxol compounds are prescribed. Vinka alkaloids have been reported to precipitate severe acute neuropathy, Guillain–Barrè-like, in previously undiagnosed CMT patients, and their use should be avoided if possible unless there is no real alternative [41]. For platin and taxol compounds, the choice depends on the risk as compared to other chemotherapy lines, and, if administered, the patients should be evaluated clinically and electrophysiologically before treatment and monitored during treatment, keeping into account the coast phenomenon of neuropathy worsening after drug discontinuation [41,42].

Genetic diagnosis is important for proper prognostic advice and for genetic counselling. Prenatal diagnosis is possible in several countries as well as preimplantation genetic diagnosis during in vitro fertilisation procedures [43].

CMT women have only a little increase in risk of pregnancy complications, such as preterm delivery, abnormal presentation, and placenta previa, with no significant increased risks for the newborns [44]; however, subjective worsening of CMT has been reported, independently from CMT type, in a percentage of cases ranging from 16% of the patients [44] to 38% of the pregnancies [45] with full recovery occurring only in a minority of cases.

## 3. Future Perspectives

While there is still no disease-modifying therapy available for any type of CMT, there are several compounds under evaluation and at different stages of advancement in the pathway to use for patients. They are summarised in Table 1.

### 3.1. PMP22 Downregulation in CMT1A

The majority of studies are focused on CMT1A, as it is the most frequent subtype, and downregulation of PMP22 overexpression is theoretically achievable with different approaches. The first trials with ascorbic acid, which are supposed to reduce PMP22 expression through a cAMP-mediated mechanism [46], did not result in any benefit [11,47]; however, they paved the way to clinical trial readiness by accelerating the development of clinical and paraclinical outcome measures, natural history studies, and multicentre multinational clinical trial design. Anti-progesterone drugs are able to decrease PMP22 overexpression in CMT1A rats with clinical and histological improvement, but risks of side effects counterbalance their hypothetical advantage [48,49]; a phase II trial with ulapristil has been carried out in France but did not recruit the target number of patients, and results have not been published (ClinicalTrials.gov NCT02600286).

PXT3003 is a mixture of low doses of baclofen, sorbitol, and naltrexone; it is predicted to lower PMP22 levels, which produced some benefit in CMT1A rodent models [50] and in a phase II trial [51]. The phase III trial results were hampered by the unblinding for about one-half of the high-dose group patients caused by crystallisation of the more concentrated mixture (ClinicalTrials.gov NCT03023540); a novel international phase III trial required by the FDA has just started. This is currently the treatment placed in the most advanced stage of development for CMT.

### 3.2. Gene Silencing and Gene Therapy

Partial gene silencing of *PMP22* with improvement of neuropathy features has been obtained in CMT1A rodent models by means of different approaches (Figure 1): (a) subcutaneous antisense oligonucleotides (ASOs) [52]; (b) subcutaneous small interfering RNA (siRNA) conjugated to squalene nanoparticles [53]; (c) intranerve injection of small hairpin RNA (shRNAs) with an adeno-associated viral serotype 9 (AAV2/9) vector [54]; and (d) injection into the sciatic nerve of liposome-encapsulated single guide RNA (sgRNA) aimed at deleting the *PMP22* TATA-Box by CRISPR-Cas9 editing [55]. Translation of these promising approaches of *PMP22* gene silencing to clinical trials is planned. There are problems of targeting a sufficient number of Schwann cells, obtaining a proper quantitative silencing to avoid the risk of developing HNPP features associated with *PMP22* underexpression, and cost–benefit ratios for a long-life therapy. Selective siRNA-mediated gene silencing of the mutated L16P *PMP22* allele was also successfully obtained in the Trembler J mouse [56]. Gene editing of mutated alleles with the CRISPR-Cas9 or similar novel techniques is also becoming feasible.

Gene therapy to insert a novel gene or to substitute a defective one is another approach that is close to entering the clinical phase in the CMT field (Figure 1).

An open label trial (NCT03520751) is planned, with injection into leg muscles of three CMT1A subjects of ascending doses of the neurotrophin 3 (NT-3) gene employing an adeno-associated viral (AAV) vector. NT-3 is a growth factor promoting nerve regeneration and remyelination, and previous experiments in mouse models of CMT1A and CMTX1 (Cx32 knockout) were successful [57,58]. The NT-3 story started with a pilot study on eight CMT1A patients treated with injections of recombinant NT-3: nerve biopsies revealed an increase in myelinated fibre density, and clinical evaluation showed decreased neurologic impairment and improved sensory scores as compared to placebo-treated controls [57].

Research is ongoing to replace CMT genes altered by loss-of-function mutations, particularly in recessive CMT but also in CMTX1. Kleopa et al. are pursuing intraneural or intrathecal approaches of gene delivery using lentivirus or AAV9 vectors and a myelin-specific *MPZ* promoter to target Schwann cells; they treated GJB1-knocked-out mice lacking Cx32 expression and the Sh3tc2^−/−^ mouse model of CMT4C, and they could observe stable protein expression in Schwann cells of nerve roots and proximal peripheral nerves and improvement of clinical and paraclinical outcomes in both models [59,60,61].

### 3.3. Neuregulin Pathway Modulation

The Neuregulin pathway is fundamental to regulate myelin thickness. In particular, Neuregulin-1 type III (Nrg1-III), produced by axons, activates the PI3K–Akt and MAPK/Erk signalling pathways in the Schwann cell and favours myelination. The BACE1 and TACE secretases are Nrg1-III regulators with opposite actions, the first by enhancing its activity and thus increasing myelin thickness, whereas the latter has an inhibitory effect (Figure 2) [62]. Niacin-Niaspan, which is commercially available, is able to increase the TACE activity and thus was employed to treat two neuropathies with hypermyelination: the autosomal recessive CMT4B1, which is characterised by an overproduction of myelin leading to the typical myelin outfolding figures, and HNPP, which is also named tomaculous neuropathy because of the sausage-shaped hypermyelinating formations called tomacula. Treatment resulted in improved nerve histology in both animal models [63]. Conversely, both genetic overexpression of Nrg1-III and the functionally equivalent pharmacological suppression of TACE were able to improve the neurophysiological and morphological phenotype of the hypo/dysmyelinating CMT1B mouse model [64]. Moreover, the soluble Neuregulin-1 type I, which is able to promote remyelination following nerve injury, improved motor performances and sciatic nerve histology in CMT1A rats [65]. Therefore, the modulation of Nrg1-type I and III is a promising approach to treat both hypo-demyelinating and hypermyelinating neuropathies.

### 3.4. Endoplasmic Reticulum (ER) Stress and Unfolded Protein Response (UPR) Activation

UPR is an adaptive, protective cellular reaction that is able to relieve stress from misfolded proteins and is activated when mutated proteins are retained in the ER. This is what has convincingly been demonstrated to occur in CMT1B for many MPZ mutations [66]. However, when Schwann cells are in chronic stress in the prolonged neuropathic process, the UPR becomes inadequate, and apoptosis, cell death, and abnormal signalling occur. Prolonged UPR activation obtained with curcumin and more recently with sephin-1 was effective in ameliorating the motor and morphological abnormalities of two CMT1B models [67,68]. Curcumin proved also effective in treating the CMT1E (Trembler-J) mouse model, which showed UPR activation consequent to ER retention of mutant PMP22 [69]. PMP22 aggregation has been shown to occur in CMT1A models and in patients’ cells [70,71], and it is possible that UPR modulation can be beneficial for treating CMT1A as well [72]. Clinical trials with sephin-1 (IFB-088), which inhibits the dephosphorylation of the eIF2a kinase in the PERK arm of UPR, are under development.

### 3.5. Axonal Degeneration and Axonal Transport

Both primary demyelinating and axonal CMT cause clinical disability through axonal degeneration. Sterile alpha and toll/interleukin 1 receptor motif-containing 1 (SARM1) is a key molecule in the axonal degeneration process, and loss of SARM1 is able to prevent axonal degeneration and improve recovery after nerve injury and in different neuropathic models [73]. Therefore, great efforts are devoted to developing SARM1 inhibitors as potential treatment of many neuropathies including CMT.

Disturbances of axonal transport are common to many hereditary neuropathies, and another active field of research is the development of selective inhibitors of Histon De-ACetylase type 6 (HDAC6), which by increasing acetylated alpha-tubulin were able to improve axonal transport and ameliorate the phenotype of mutant HSPB1 mice, which is a model of CMT2/dHMN [74]. Thereafter, the inhibition or deletion of HDAC6 have been tested in other two neuropathy models, the mutant Gars^C201R/+^ mice (model of CMT2D/dHMN type VA) and the MFN2R94Q mouse model of CMT2A, with effective correction or prevention of electrophysiological and clinical dysfunction [75,76]. An improvement of mitochondrial transport has been shown in HSPB1 and GARS models [74,75] and is likely to be a fundamental component of the beneficial action of HDAC6 inhibitors, restoring energetic metabolism; MFN2 is important for mitochondrial fusion but also for mitochondrial transport, and HDAC6 inhibition improved the phenotype of the MFN2R94Q mouse [76], suggesting that mitochondrial trafficking is an important target for therapy even when mitochondria are dysfunctional. Whether this approach is valid for all CMT2 types or even for all axonal neuropathies is still unknown, but HDAC6 inhibitors are currently considered potentially effective treatment for axonal neuropathies worthy of further exploration and testing in clinical trials.

### 3.6. Other Approaches for Demyelinating CMT1

There are several compounds that have been tested or are under consideration for different CMT models and types, and the most interesting are listed below.

A phase II trial (ClinicalTrials.gov NCT03124459) has been recently completed that employed ACE-083, which is a molecule that is able to generate muscle hypertrophy by inhibiting myostatin and other muscle regulators [77]. A total of 62 CMT1 and CMTX1 patients were treated with periodic injections of ACE-083 into the tibialis anterior muscles, which indeed produced a muscle volume increase; however, this did not translate into a significant amelioration in functional and quality of life measures as compared to sham-treated patients. Therefore, the planned phase III trial was abandoned.

Studies on CMT1A rats have shown that PMP22 overexpression results in increased expression of the P2X 7 purinocereptor, which is an ion channel involved in several functions, leading in turn to a potentially deleterious increased Ca^2+^ concentration in Schwann cells [78]. P2X7 silencing or use of its antagonists restored proper Ca^2+^ concentration in CMT1A SC, and the P2X7 receptor inhibitor A438079 improved the clinical and histological phenotype of CMT1A rats [79]. The favourable safety profile shown by A438079 in preceding trials in rheumatoid arthritis encourages its use in clinical trials for CMT1A [80].

Another active field of research concerns the lipid metabolism, which is altered at least in CMT1A, and the consequent abnormalities in lipid composition of myelin and membranes. The supplementation of phosphatidylcholine and phosphatidylethanolamine with the diet was beneficial for CMT1A rats both biochemically and clinically [81]. Therefore, a clinical trial is being planned with oral lecithin for CMT1A patients.

Some studies have revealed in the MPZ knockout mice, a model of dysmyelination, an abnormal expression in motor axons of the Na_V_1.8 isoform of the sodium channel, which is specific to the sensory neurons [82]; in further experiments, Na_V_1.8 blocker treatment reverted the progressive motor impairment in P0-deficient mice [83]. Thus, sodium channel blockers might be a potential treatment for severe dysmyelinating CMT [84].

There is evidence for a contribution of inflammation to pathomechanisms of at least some CMT forms: studies from CMT1A, CMT1B, and CMTX1 mouse models reveal that phagocytising macrophages cause further nerve damage. The inhibition of the colony-stimulating factor 1 (CSF1), which activates macrophages, decreased their presence in the nerves and improved clinical and pathological features in CMT1B and CMTX1 mice [85].

### 3.7. Correction of Metabolic Abnormalities

Some CMT types represent true metabolic neuropathies resulting from enzymatic deficits leading to product shortage or neurotoxic accumulation; in such cases, correction of the metabolic anomalies can potentially rescue the neuropathic phenotype. This is the case for CMT4B1 and CMT4B2, the hypermyelinating neuropathies already mentioned above, which are caused by mutations in myotubularin-related proteins (MTMR2 and MTMR13, respectively) involved in phosphoinositides’ metabolism and for which inhibitors of the PIKfyve kinase, the enzyme counteracting the activity of MTMR2, are under investigation as potential treatments [86].

A second example is provided by mutations in the two genes (*SPTCL1/2*) coding for serine palmitoyltransferase (SPT), which is the first enzyme in the sphingolipid synthesis pathway that conjugates L-serine with palmitoyl-CoA. Such *SPTCL1/2* mutations lead to an accumulation of neurotoxic 1-deoxysphingolipids where serine is substituted by alanine or glycine and causes Hereditary Sensory Neuropathy type 1 (HSN1); L-serine supplementation was metabolically and phenotypically effective in the mouse model and improved the metabolic abnormalities also in the patients, although a pilot trial did not result in significant clinical benefit, which was likely because it was underpowered [87].

The X-linked CMTX5 and two related allelic disorders, Arts syndrome and the X-linked nonsyndromic sensorineural deafness DFNX1, are associated with mutations in an enzyme involved in purine metabolism, phosphoribosylpyrophosphate synthetase 1 (PRPS1). Two patients with Arts syndrome appeared to benefit from S-adenosylmethionine (SAM) supplementation administered with the aim to refill the purine nucleotides pool [88]; whether SAM is effective for CMTX5 is still unknown.

The most recent and interesting instance is constituted by the recessive mutations in the gene encoding the sorbitol dehydrogenase (SORD) enzyme, associated with CMT2/dHMN, which catalyses the first step of the polyol pathway by oxidising sorbitol into fructose. SORD-defective patients have very high levels of sorbitol in their blood; studies on diabetic neuropathies have shown that sorbitol may be toxic for the nerves and inhibitors of aldose reductase, which transforms glucose into sorbitol in the step preceding SORD action, can reduce sorbitol accumulation and were effective in the SORD-defective cellular models [89].

## 4. Conclusions

Nowadays, the main challenge for CMT is to find disease-modifying treatments. Although no drug treatment for any CMT type is still available, in the last few years, great advances have been made: the first clinical trials in patients have been conducted, others are ongoing or planned, and several therapeutic approaches are being tested in experimental models. There is a cautious optimism among researchers that it will not take a long time to see the first effective therapies for CMT. In the meanwhile, there is a lot we can do for CMT patients, including proper rehabilitation therapies, the use of appropriate orthotics, surgery for skeletal deformities in selected cases, management of pain, cramps, and fatigue, adequate general care, avoidance of neurotoxic drugs, and genetic counselling.

## Figures and Tables

**Figure 1 brainsci-11-01447-f001:**
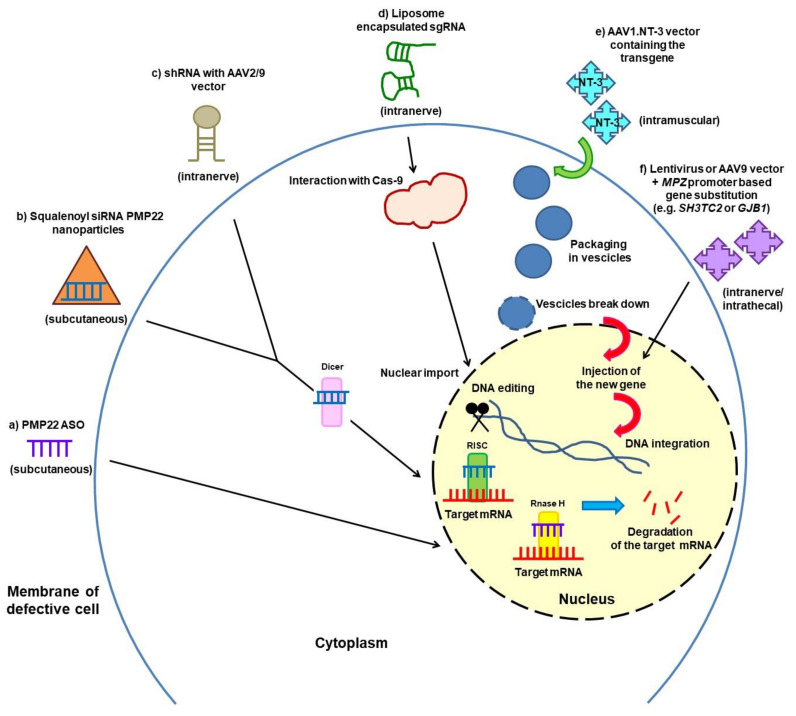
Partial silencing of *PMP22* gene (a–d) and gene therapy against CMT1A and CMTX1 (e,f) in different rodent models. (a) ASOs decrease *PMP22* mRNA in affected nerves in two CMT1A mouse models. (b) siRNAs conjugated to squalene nanoparticles normalise Pmp22 protein levels in two CMT1A mouse models. (c) AAV serotype 9 viral vectors expressing shRNA directed against *PMP22* mRNA restore wild-type expression levels of PMP22 in CMT1A rats. (d) Liposome encapsulated sgRNAs delete the *PMP22* TATA-box by CRISPR-Cas9 editing in a mouse model of CMT1A. (e) AAV1.NT-3 gene therapy in transgenic mouse models (TremblerJ and Cx32 knockout mice) and planned in three CMT1A patients. (f) Gene delivery using lentivirus or AAV9 vectors and a myelin-specific *MPZ* promoter to target murine Schwann cells (Sh3tc2^−/−^ and Cx32 knockout mice). For each approach, the route of administration is also indicated in parentheses.

**Figure 2 brainsci-11-01447-f002:**
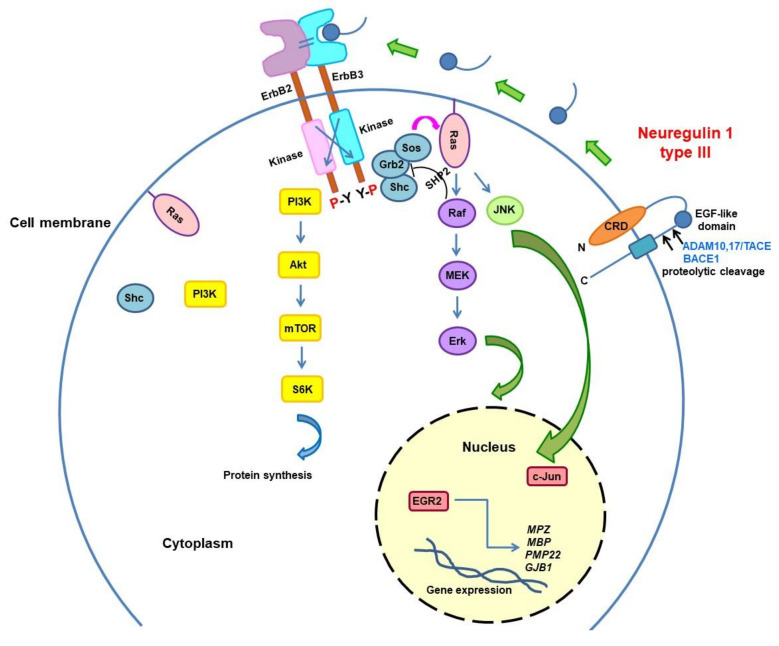
Neuregulin1/ErbB signalling determines myelin sheath thickness. Neuregulin1 type III (Nrg1-III) has an epidermal growth factor-like (EGF-like) domain and contains a cysteine-rich domain (CRD) embedded in the lipid bilayer, which leaves the N-terminal side tethered to the membrane. Nrg1-III, produced by axons, is proteolytically cleaved by proteases of the ADAM family and BACE1. The BACE1 and TACE (also known as ADAM17) secretases are Nrg1-III regulators with opposite actions, the first by enhancing its activity and thus increasing myelin thickness, whereas the latter has an inhibitory effect. Nrg1-III provides the ligand for ErbB3 receptor leading to its heterodimerisation with ErbB2, activation of the tyrosine kinase domain, and phosphorylation of the cytoplasmic region of the ErbB partner. This event causes various adaptors/effectors’ (e.g., Ras-Shc, PI3K) recruitment and activation of multiple intracellular signalling pathways such as the PI3K–Akt and MAPK/Erk pathways in the Schwann cell.

**Table 1 brainsci-11-01447-t001:** Main therapeutic approaches and compounds under study for Charcot-Marie-Tooth disease and related neuropathies.

CMT Type	Compound/Approach	Mechanism of Action/Aim of Therapy	Clinical Trials Status and Comments
	Ascorbic acid	Reduction of *PMP22* expression by reducing cAMP levels	Phase III studies concluded; all failed to meet their primary endpoints and did not show a significant effect
CMT1A	Progesterone antagonists/modulators: onapristone, ulapristal	Reduction of PMP22 synthesis	Onapristone: unacceptable side effects. Ulapristal: phase II trial conducted (*n* = 23 out of 45 planned). Results not available
	PXT3003 (mix of low doses of baclofen, sorbitol and naltrexone)	Inhibition of SCs proliferation and reduction of the synthesis of PMP22; baclofen, GABA_B_ receptor modulator	PXT3003: phase II (*n* = 80) concluded. Phase III (*n* = 323) concluded but unblinding problems in the high-dose group. New Phase III requested by FDA and just started
CMT1A, CMT1E	Gene silencing (ASOs, siRNAs, shRNAs, sgRNAs—CRISPR/Cas9)	Partial silencing of overexpressed (CMT1A) or mutated (CMT1E) *PMP22*	Under consideration. Issues: targeting a sufficient number of SCs, obtaining a proper quantitative silencing to avoid risk of HNPP, long-life therapy
CMT1A, CMTX1, CMT4C	Gene therapy (e.g., AAV1-NT-3; *GJB1* and *SH3TC2* gene substitution)	Gene insertion (*NT-3* = neurotrophic action) or (*GJB1*, *SH3TC2*) substitution	NT-3: open trial (*n* = 3) planned for CMT1A Gene substitution: still in preclinical phases
CMT1A, CMT1B, CMT4B, HNPP	Neuregulin pathways (particularly Neuregulin-1 III)	Regulation of myelin thickness	Niacin-niaspan candidate for CMT4B and HNPP?
CMT1A, CMT1E, CMT1B	Curcumin, sephin-1	UPR inhibition by attenuation of the IRE1 branch	Possible clinical trial in CMT1A/CMT1B
CMT	FLX-787	Activation of TRPA1 and TRPV1 channels, for cramps	Phase II (*n* = 120) stopped for oral intolerability in a subset of patients
All CMT and related neuropathies	SARM1 inhibitors	Prevention of axonal degeneration	--
CMT2F, dHMN2A, CMT2A, CMT2D, dHMN5 (and others?)	HDAC6 inhibitors	Reduction of microtubules acetylation, improvement of axonal transport	Under consideration
CMT1, CMTX1	ACE-083	Action on myostatin pathway	Phase I+II (*n* = 62 overall) trial did not produce significant clinical improvement. Planned phase III trial abandoned
CMT1A	P2X7 receptor modulators (e.g., A438079)	Reduction of abnormal calcium influx into SCs	P2X 7 antagonist acceptable safety and tolerability in a previous phase II trial in rheumatoid arthritis
CMT1A, other CMT?	Dietary lipid supplementation	Dietary correction of defective myelin lipid biosynthesis	Trial with oral lecithin supplementation planned in Germany
CMT1B, other dysmyelinating CMT?	Sodium channel blockers	Blocking of Nav 1.8 channel	Lamotrigine could be a candidate compound
CMT1A, CMT1B, CMTX1	CSF1R inhibitors	Decreased number and activity of macrophages in the nerve	--
CMT4B1, CMT4B2	PIKfyve enzyme inhibitors	Inhibition of PIKfyve and decrease of PI3,5P2 levels	To be considered for CMT4B1 and CMT4B2
HSN1	L-Serine	Reduction of neurotoxic deoxysphingolipids	Phase II trial (*n* = 18) performed, primary endpoint not reached, but underpowered trial
CMTX5 and allelic disorders (Arts syndrome and DFNX1)	S-adenosylmethionine (SAM)	Purine nucleotides supply	Anecdotal report
CMT and dHMN associated with biallelic SORD mutations	Aldose reductase inhibitors	Inhibition of aldose reductase, the enzyme converting glucose into sorbitol	Under consideration

AAV1-NT-3 = adeno-associated virus-mediated neurotrophin-3, ASOs = antisense oligonucleotide, cAMP = cyclic adenosine monophosphate, CMT = Charcot-Marie-Tooth disease, CRISPR/Cas9 = clustered regularly interspaced short palindromic repeats/CRISPR-associated protein 9, CSF1R = colony-stimulating factor 1 receptor, DFNX1 = X-linked deafness-1, FDA = Food and Drug Administration, GABA_B_ = gamma-aminobutyric acid B receptor, HDAC6 = histone deacetylase 6, dHMN = distal hereditary motor neuropathy, HNPP = hereditary neuropathy with liability to pressure palsies, HSN = hereditary sensory neuropathy, IRE1 = inositol-requiring enzyme 1, PIKfyve = phosphatidylinositol 3-phosphate 5-kinase, SARM1 = sterile alpha and TIR motif containing 1, SCs = Schwann cells, sgRNAs = single guide RNAs, shRNAs = short hairpin RNAs, siRNAs = small interfering RNAs, TRPA1 = transient receptor potential cation channel, subfamily A, member 1, TRPV1 = transient receptor potential cation channel subfamily V member 1, UPR = unfolded protein response.

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
