# Peer review of "Challenges in Treating Charcot-Marie-Tooth Disease and Related Neuropathies: Current Management and Future Perspectives"

_brainsci, 2021, doi:10.3390/brainsci11111447_

Round 1
Reviewer 1 Report
This review is well written and covers important things about the latest CMT treatments.
It also describes rehabilitation and gives useful information to the doctor in charge.
In particular, there is nothing to change.
Author Response
Reviewer 1: This review is well written and covers important things about the latest CMT treatments. It also describes rehabilitation and gives useful information to the doctor in charge. In particular, there is nothing to change.
Response: We thank the reviewer for the comment.
Reviewer 2 Report
The authors in this review titled “Challenges in treating Charcot-Marie-Tooth disease and related neuropathies: current management and future perspectives”” give a good account of the genetic bases of peripheral neuropathies as well as how to alleviate pain, fatigue and other related symptoms in patients. The review also highlights the main therapeutic approaches and compounds tested or under study for CMT and other related neuropathies.
The review reads generally well, is of potential interest to a broad audience interested in peripheral development and neuropathy. It is organised in a logical way and key information is represented in a table and two figures.
I have no major concern regarding the manuscript.
I have two minor issues that I hope could help in improving the manuscript:
- The HDAC6 inhibitors strategy is quite intriguing. Is it valid for all sort of axonal related CMT2 neuropathies? In other words, are we able to improve nerve function just by enhancing axonal transport even if mitochondrial function per se is defective? How are these two related? Is it possible to discuss this further?
- Might need to elaborate more on Figure 2. There is no information regarding the Ras-Shc and PI3K proteins sitting on the left side of the cell.
Author Response
Reviewer 2: The authors in this review titled “Challenges in treating Charcot-Marie-Tooth disease and related neuropathies: current management and future perspectives”” give a good account of the genetic bases of peripheral neuropathies as well as how to alleviate pain, fatigue and other related symptoms in patients. The review also highlights the main therapeutic approaches and compounds tested or under study for CMT and other related neuropathies.
The review reads generally well, is of potential interest to a broad audience interested in peripheral development and neuropathy. It is organised in a logical way and key information is represented in a table and two figures.
I have no major concern regarding the manuscript.
I have two minor issues that I hope could help in improving the manuscript:
The HDAC6 inhibitors strategy is quite intriguing. Is it valid for all sort of axonal related CMT2 neuropathies? In other words, are we able to improve nerve function just by enhancing axonal transport even if mitochondrial function per se is defective? How are these two related? Is it possible to discuss this further?Might need to elaborate more on Figure 2. There is no information regarding the Ras-Shc and PI3K proteins sitting on the left side of the cell.
Response: We thank the reviewer for the valuable and useful comments and suggestions. We have revised the manuscript accordingly and below you will find the list of point-to-point responses and related changes.
Point 1: We have added on page 8, line 304 the following text: “Improvement of mitochondrial transport has been shown in HSPB1 and GARS models [75,76] and is likely to be a fundamental component of the beneficial action of HDAC6 inhibitors, restoring energetic metabolism; MFN2 is important for mitochondrial fusion but also for mitochondrial transport and HDAC6 inhibition improved the phenotype of the MFN2R94Q mouse [77], suggesting that mitochondrial trafficking is an important target for therapy even when mitochondria are dysfunctional. Whether this approach is valid for all CMT2 types or even for all axonal neuropathies is still unknown, but”
Point 2: We have added in the figure legend on page 12, line 412 the following text: “activation of the tyrosine kinase domain and phosphorylation of the cytoplasmic region of the ErbB partner. This event causes various adaptors/effectors' (such as Ras-Shc, PI3K) recruitment and”